# Measuring the Contrast Sensitivity Function in Non-Neovascular and Neovascular Age-Related Macular Degeneration: The Quantitative Contrast Sensitivity Function Test

**DOI:** 10.3390/jcm10132768

**Published:** 2021-06-24

**Authors:** Filippos Vingopoulos, Karen M. Wai, Raviv Katz, Demetrios G. Vavvas, Leo A. Kim, John B. Miller

**Affiliations:** 1Harvard Retinal Imaging Lab, Massachusetts Eye and Ear, Harvard Medical School, 243 Charles St., Boston, MA 02114, USA; Filippos_Vingopoulos@meei.harvard.edu (F.V.); Raviv_Katz@meei.harvard.edu (R.K.); 2Retina Service, Massachusetts Eye and Ear, Department of Ophthalmology, Harvard Medical School, 243 Charles St., Boston, MA 02114, USA; Karen_Wai@meei.harvard.edu (K.M.W.); Demetrios_Vavvas@meei.harvard.edu (D.G.V.); leo_kim@meei.harvard.edu (L.A.K.)

**Keywords:** contrast sensitivity function, visual function metrics, qCSF, age-related macular degeneration, contrast sensitivity

## Abstract

Age-related macular degeneration (AMD) affects various aspects of visual function compromising patients’ functional vision and quality of life. Compared to visual acuity, contrast sensitivity correlates better with vision-related quality of life and subjectively perceived visual impairment. It may also be affected earlier in the course of AMD than visual acuity. However, lengthy testing times, coarse sampling and resolution, and poor test–retest reliability of the existing contrast testing methods have limited its widespread adoption into routine clinical practice. Using active learning principles, the qCSF can efficiently measure contrast sensitivity across multiple spatial frequencies with both high sensitivity in detecting subtle changes in visual function and robust test–retest reliability, emerging as a promising visual function endpoint in AMD both in clinical practice and future clinical trials.

## 1. Introduction

Age-related macular degeneration (AMD) is currently the leading cause of visual impairment in subjects older than 50 years of age in developed countries and the third-leading cause worldwide [1]. Given the anticipated aging of the global population, the prevalence of AMD is expected to increase from approximately 30–50 million people worldwide in 2017 [2] to 288 million by the year 2040 [1]. Severe reductions in visual acuity (VA) typically manifest late in the course of the disease, limiting the clinical utility of the traditional visual acuity test in early and intermediate stages of AMD [3,4,5]. Even with relatively high visual acuity, it is not uncommon for patients with early or intermediate AMD to complain of subjective visual impairment and demonstrate reduced vision-related quality of life (VRQoL) scores [3,5,6,7].

### 1.1. Limitations of Visual Acuity Testing

The current gold standard for the evaluation of visual function in the clinical practice dates back to 1862, when Herman Snellen developed a standard letter chart that has widely remained in use with only minor modifications ever since. By showing progressively smaller letters at a given distance and at high contrast, the Snellen chart assesses the minimal angle of resolution (the smallest letters resolved) which defines visual acuity, the visual system’s resolving power. Similarly, the Early Treatment Diabetic Retinopathy Study (ETDRS) chart measures the minimal angle of resolution to define VA, but in contrast to the Snellen chart, it uses a regular geometric progression of the size and spacing of the presented letters, employing a logarithmic scale with increments of 0.1 log units. The scoring method for both is the line assignment method, where a patient gets credit for lines, not letters read. Partially read lines are given a ‘ + ’ or ‘ − ’ designation, though there is no accepted standard on how this is recorded.

Although VA is the traditionally used functional outcome metric both in the routine clinical setting and in clinical trials, it seems that it may not be the ideal test, especially in AMD. VA is a relatively crude test that fails to reveal subtle changes in visual function (two letter loss, or logMAR 0.04) [8]. Further, the relatively high inherent variability of the test (±0.15 logMAR or 1.5 lines on a standard logMAR chart) forces clinicians to look for a difference of at least 0.15 logMAR (eight (8) letters on a standard logMAR visual acuity chart) between measurements to be confident that a real change in VA has occurred [9]. Owsley et al. found only a two-letter VA loss in early AMD compared to controls, which was statistically significant yet of little clinical significance, given the greater margin of variability for VA testing [3].

Further, VA is tested with the ETDRS chart in clinical trials versus the Snellen chart in clinical practice, with translational discrepancies being inherently linked to how VA is currently tested. A recent study looking at VA differences between the two widely used charts in subjects entering clinical trials, reported a significant VA variability of 0.14 logMAR (6.7 letters) in both non-neovascular (dry) and neovascular (wet) AMD and progressively increased VA variability was observed with progressively worse VA [10].

### 1.2. The Value of Contrast Sensitivity Testing in AMD

Despite the well-known limitations of using VA to characterize early functional deficits or quantify disease progression, VA remains the most widely accepted and used metric in evaluating visual function in AMD. To better assess visual function in AMD, various alternative metrics of functional outcome have been investigated, including dark adaptation [11], microperimetry [12,13,14], multifocal ERG [15], photo-stress recovery time [16], cone-mediated flicker sensitivity [17], photopic or scotopic light sensitivity [18,19], the Moorfields Acuity Charts [20] and contrast sensitivity [21,22,23]. Among these, contrast sensitivity (1) seems to correlate better with subjective visual impairment and vision-related quality of life compared to VA [24,25,26,27], (2) may be affected earlier in the course of neurodegenerative disorders [28,29,30] and (3) may detect more subtle changes in visual function [24,31,32]. Contrast sensitivity quantifies the lightness or darkness needed to identify a target against its background. Chromatic contrast sensitivity will not be discussed herein, as it describes the ability to discriminate between stimuli based on their chromaticity difference alone, independent of any luminance contrast.

Perhaps the most important pre-requisite of any functional outcome metric is its patient-relevance. This can be assessed by the correlation of the functional metric with functional vision, which is approximated by vision-related quality of life (VRQoL) questionnaires [33,34]. Pondorfer et al. reported that contrast sensitivity in dry AMD had a stronger effect on VRQoL compared to VA in all three subscales of the Impact of Vision Impairment questionnaire including reading, mobility and emotional well-being [24]. Further, Roh et al. reported that contrast sensitivity is an important factor affecting VRQoL in patients with vision impairment due to bilateral advanced AMD [35]. In the large cohort of 3654 patients reported by the Blue Mountains Eye Study, contrast sensitivity was strongly associated with all self-reported measures of visual impairment such as trouble driving at night, facial recognition, reading, and watching TV, more so than VA [25]. Of note, contrast sensitivity seems to be an independent factor relative to VA for self-reported visual disability [26]. Given the strong correlation between contrast sensitivity and VRQoL, Rubin et al. reported that a six (6)-letter loss of contrast sensitivity on the Pelli-Robson chart has a similar impact on self-reported visual disability as a 15-letter loss of visual acuity [26].

Further, compared to VA, contrast sensitivity might be affected earlier in the course of neurodegenerative ocular disorders, which allows for earlier detection of deficits in visual function before the effect on VA has become apparent [28,29,30]. It is hence possible that in some cases VA underestimates the onset and severity of visual disability, overestimating the patient’s actual visual capacity [13]. In non-neovascular (dry) AMD, VA may not be the ideal functional metric to quantify disease progression, as it is often only affected late in the course of the disease [3]. In a group of 90 dry AMD patients, Pondorfer et al. found no significant differences in VA between early and intermediate AMD patients. Yet contrast sensitivity was significantly decreased in intermediate AMD relative to the early AMD group, suggesting that it may be a useful adjunct visual function metric when evaluating disease progression [24]. Not only can contrast sensitivity detect subtle visual function deficits early in the disease course and measure subtle deteriorations in disease progression, but it may also be able to reveal subtle improvements following therapeutic intervention, which may otherwise go unnoticed by measuring VA alone [31,32].

A functional outcome metric more sensitive than VA to detect subtle changes in visual function following treatment would also be able to detect smaller differences between different treatment arms, which in turn would mean that smaller sample sizes would be required for clinical trials to be adequately powered. Contrast sensitivity has shown promise for measuring the progression of vision loss in eye diseases or their remediation with treatment. Importantly, contrast sensitivity has been able to detect anti-VEGF treatment effects on visual function in cases where VA improvement seems to have stalled [31,32]. Bansback et al. found a strong relationship between contrast sensitivity and health-related quality of life in AMD, suggesting that evaluating VA alone might underestimate the benefits of ocular treatments in AMD patients [36]. Further, a review by Mones et al. suggested that the use of contrast sensitivity as an outcome measure in nAMD clinical trials may be a better predictor of activities of daily living, mobility and orientation than VA [21].

Despite its promising role in visual function assessment, the practical constraints limiting the sensitivity and/or precision of most contrast sensitivity testing methods have thus far prevented wider adoption of contrast sensitivity function in clinical practice and clinical trials.

## 2. Discussion

### 2.1. Currently Available Contrast Sensitivity Tests

#### 2.1.1. Conventional Laboratory Contrast Sensitivity Testing

Conventional laboratory testing can measure contrast sensitivity function over an expansive range of spatial frequencies and contrast levels combinations [37]. Yet, under typical designs, adding a spatial frequency condition requires a minimum of 50–100 experimental trials, therefore sampling at 5–10 spatial frequencies typically requires 500–1000 trials [37,38]. Consequently, employing conventional laboratory testing to characterize contrast sensitivity function in a single patient can take up to an hour, rendering this method too time consuming and thus impractical in the clinical setting [37].

Due to severe testing time constraints, clinical contrast sensitivity tests are far more condensed (and less precise) than laboratory testing. As early as 1977, Sjostrand and Frisen used sinusoidal gratings of variable contrasts and spatial frequencies on a television-based display to test patients with macular disease versus controls [39]. Since then, there have been multiple different commercially available tests to measure contrast sensitivity function in the clinical setting; for the purposes of this review, we focus on the most highly utilized tests in AMD.

#### 2.1.2. The Pelli-Robson Chart 

First introduced in 1988, the Pelli-Robson contrast sensitivity letter chart is the most widely used test to measure contrast sensitivity [40]. All letters are of the same optotype size (4.8 cm), and at the testing distance of 1 m which the test’s manual suggests, this corresponds to a Snellen equivalent of approximately 20/650. Hence, employing the Pelli-Robson test, clinicians only measure contrast sensitivity at one spatial frequency, which in the typical clinical setting is between 0.5–1 cycles per degree. Across each line of three letters, the contrast level decreases by 0.15 log unit. Per testing protocol, patients must identify two of the three letters correctly, at the manufacturer’s specified testing distance of 1 m, in order to get credit for that line. The Pelli-Robson chart tests contrast sensitivity thresholds from 100% to 0.56%, while overall being a quick, easily applied, inexpensive test to perform with good test–retest repeatability [41,42].

As in most retinal disorders, the Pelli-Robson is the most commonly employed contrast sensitivity test in AMD studies. In the non-neovascular (dry) form, Roh et al. and Ponderfer et al. used the Pelli-Robson chart to measure contrast sensitivity and reported associations with quality of life in AMD patients [24,35]. Treatment effects have also been demonstrated with the Pelli-Robson chart: Nixon et al. showed improvement in contrast sensitivity as measured with Pelli Robson chart in neovascular AMD patients injected with aflibercept after being switched from ranibizumab [22]. Neely et al. and Owsley et al. found no difference in contrast sensitivity between eyes with early AMD compared to healthy controls, using the Pelli-Robson chart [3,23]. Given that the Pelli-Robson chart is a coarse method that only measures contrast sensitivity in one spatial frequency (0.5–1 cycles per degree), it is possible that deficits in contrast sensitivity are missed if there are reductions at intermediate and higher spatial frequencies [41]. Further, coefficients of repeatability (1.96 times standard deviation of test-retest differences) range from 0.13 to 0.48 log10 units and because of the coarse quantization of the Pelli-Robson chart, a change in test score of at least ±0.3 or even 0.45 log10 for low-vision patients, is required to detect a significant change [43]. Further, as with testing using any other chart, there are technical issues such as difficulties with maintaining consistent illumination and reflections from the surface of the chart, and this is nearly impossible to control among different testing room environments. 

#### 2.1.3. Vistech Testing and Related Charts

In order to measure contrast sensitivity in different spatial frequencies, several tests have been introduced, yet for portability and ease of application these tests are pre-printed letter charts with pre-determined, hence a priori limited, frequencies and contrast levels [44].

The Vistech Vision Contrast Test System test was first introduced by Ginsburg in 1984 [45]. The test contains circular photographic plates of sine wave gratings in six rows with different spatial frequencies and nine columns of varying contrast levels. The sine wave gratings are tilted in different orientations and the patient has to identify the orientation of each grating. There have been various modified versions of this chart including VCTS-6500 and MCT-8000, which allow the option for the observer to pick “blank” as an option if they do not see the orientation of the gratings. The step size of contrast in each column is irregular, but the average step size is about 0.23 log unit steps [42]. 

Kleiner et al. applied Vistech testing in a small cohort of dry AMD patients, demonstrating a loss of contrast sensitivity at high spatial frequencies relative to normal controls [28]. The CSV-1000 chart is another similar pre-printed grating chart to test contrast sensitivity (VectorVision Dayton, OH, USA) [46]. It utilizes internal retro-illumination to decrease issues with uneven lighting in different testing environments and operates at four different spatial frequencies testing nine different contrast levels at each frequency. The contrast level ranges from 0.5% to 67%, with a decrease of about 0.16 log unit between each level. However, this test shows two patches at each contrast level and the observer is asked to identify the one with contrast gratings, allowing for a 50% chance of guessing correctly [42].

The population-based Blue Mountains Eye Study used the CSV-1000 chart to measure contrast sensitivity yet there were no normal values for reference available and the authors mention that they used this test because it was the ‘simplest test available at the time’ and because ‘there were no practical, well-validated alternatives’ [25].The Functional Acuity Contrast Test is considered the second-generation modification of the Vistech test [47]. Similar to the Vistech, it utilizes circular photographic plates arranged in five rows and nine columns, but smaller regular step sizes of 0.15 log units, resulting in an overall range of 1.2 log units.

All these pre-printed grating charts measure contrast sensitivity in different spatial frequencies and do not require the patient to have knowledge of the Roman alphabet. Yet, repeated studies have shown these charts have poor test–retest reliability [48,49,50,51,52], especially at lower spatial frequencies, making them non-ideal for clinical use [41]. Further, they offer a limited number of contrast-frequency combinations, thus only test a limited range and resolution of stimuli [53,54]. In the FACT test there is also a ceiling effect as the number of columns is not increased, but smaller step sizes are used [54,55]. Last, as observers essentially have either two or three options to pick for orientation of the grating patch, it allows for a relatively high chance of choosing the correct orientation with a random guess (33% to 50% chance).

#### 2.1.4. The Spaeth/Richman Contrast Sensitivity Test 

The Spaeth/Richman Contrast Sensitivity Test (SPARCS) is a computer-based test that measures contrast sensitivity thresholds in a patient’s central and peripheral vision. Patients are asked to fixate on the central area of a grid with nine boxes while the peripheral boxes show various vertical square-wave gratings. Patients are asked to identify which area showed the vertical square wave gratings. A contrast threshold is measured for each of the tested areas that correspond to a supero-temporal, inferotemporal, inferonasal, inferotemporal, and central vision field for each eye [56].Overall test–retest reliability was reported to be 87% yet significantly lower for the peripheral vial fields. (61% to 80%) [57]. The range of tested contrast with SPARCS is from 100% to 0.45% and between each level, the contrast is decreased by about 0.15 log units. The SPARCS test is unique in its ability to test contrast thresholds in peripheral visual fields, yet similar to the Pelli Robson it only tests contrast at a singular fixed spatial frequency. Faria et al. used the SPARCS test in AMD patients and found a significantly lower score for both the central area and all four peripheral quadrants compared to control patients [57]. Since AMD predominantly affects the central vision, testing contrast thresholds in peripheral visual fields may not be of great additional value when evaluating disease progression.

### 2.2. The Quantitative Contrast Sensitivity Function (qCSF) Method

#### 2.2.1. Introduction on the qCSF Method

In 2010, Lesmes et al. introduced the quantitative contrast sensitivity function (qCSF) method [58]. This novel method employs a Bayesian active learning algorithm for measuring contract sensitivity to maximize information extraction over a very large set of possible stimuli, while at the same time reducing the number of trials, in order to reliably estimate the contrast sensitivity function from several hundred (such as in traditional laboratory methods) to several dozens. The respective time for test completion is reduced to 2–5 min per eye [58], a reasonable time that allows for a contrast sensitivity test to be integrated into routine clinical practice. 

The qCSF estimates contrast sensitivity function by presenting spatially filtered optotypes to the patient that modulate in both spatial frequency and contrast, thus enabling the efficient testing of contrast sensitivity across multiple spatial frequencies in parallel [59]. The qCSF has shown great test–retest reliability, validated in a normative dataset [60], and high sensitivity in detecting subtle changes of visual function [61].

Unlike older contrast sensitivity tests like the Pelli-Robson chart [40] that use coarse quantization and sampling, operating only in one spatial frequency [62], the qCSF measures the CSF in many different spatial frequencies, allowing for identification and measurement of disproportionate reductions in contrast sensitivity at specific spatial frequencies or global changes specific to the type of retinal disease. As discussed previously, current clinically available CSF tests that evaluate both the spatial frequency and contrast are typically pre-printed letter charts with poor range of sampling and poor test–retest reliability [42,52,55,63]. To compare with the 45 grating stimuli (five frequencies × nine contrasts) in the FACT charts, the qCSF can sample (at a minimum) a set of 720 grating stimuli (12 spatial frequencies × 60 contrasts) [58]. After application to basic studies of vision [64,65], the qCSF computational approach was commercialized in a novel clinical device, the Manifold Contrast Vision Meter (Adaptive Sensory Technology (AST), San Diego, CA, USA) [59].

#### 2.2.2. The qCSF Testing Method

The first generation of the computerized AST Manifold Platform consists of a small-form factor Intel PC with a large-format LED screen with a luminance of 95.4 cd/m^2^ and resolution of 1920 × 1080 pixels, and a tablet held by the examiner. (Figure 1A) At a viewing distance of 400 cm, the screen allows the display of optotypes in a spatial frequency range from 1.4 to 36.2 cycles per degree, which includes the whole set of frequencies mandated by the FDA (1.5 to 18 cpd) [66].

Three filtered Sloan letters of the same spatial frequency are simultaneously displayed in a horizontal line on the LED screen with decreasing contrast. The contrast of the right letter is chosen by the qCSF algorithm and is usually near threshold contrast, with the middle and left letters displayed at two and four times the contrast of the right letter, respectively [59]. (Figure 1B) The patient verbally reports the three letters presented on each screen to the examiner, who operates the test with a handheld tablet, recording “correct,” “incorrect” or “no response.” System usability scale questionnaires [67] distributed to technicians who have previously used the qCSF in the clinical setting have shown the qCSF test is almost unanimously “easy to use” [59].

The in-built adaptive Bayesian active learning algorithm uses a one-step-ahead search to identify the next grating stimulus (defined by frequency and contrast) that maximizes the expected information gain [68] in a way such that data collected at single spatial frequency improve sensitivity estimates across all frequencies. This allows for the device to select and display to each patient personalized optotypes of optimal contrast-spatial frequency combinations that are based on their previous responses. This allows for a global estimation of the contrast sensitivity function’s shape rather than local estimation at single frequencies, enabling contrast sensitivity testing over a wide range of contrast (0.2% to 100%) and spatial frequency (approximately 1 to 27 cycles per degree using a 3m viewing distance) in testing times that are feasible in the standard clinical practice. 

Using a novel active learning algorithm, the qCSF estimates the two-dimensional curve of the contrast sensitivity function, which describes thresholds as a function of spatial frequency. Only 25 trials are needed to estimate the broad, global metric of contrast sensitivity represented by the area under the logarithm of contrast sensitivity function (AULCSF), which represents CSF estimates integrated across spatial frequencies ranging from 1.5 cycle per degree (cpd) to 18 cpd [58] (Figure 2). Other outcome measures include contrast sensitivity thresholds at 1, 1.5, 3, 6, 12, and 18 cpd, which represent the least amount of contrast seen at each specific spatial frequency. Contrast acuity is also measured, which represents the smallest optotype seen at full 100%contrast: the intersection of the contrast sensitivity function curve with the x-axis. An example of the qCSF outcome measures is shown in Figure 2. 

#### 2.2.3. Current Applications of the qCSF Method in Retinal Diseases 

The qCSF method has been used in preliminary studies to measure CSF in several clinical populations including amblyopia [69,70] multiple sclerosis [71], dry age-related macular degeneration [72], glaucoma [73], retinal detachment [74], retinal vein occlusion [75,76], early diabetic retinopathy [77], and aging [78]. An initial clinical study showed that qCSF test results correlate better with subjective visual impairment and patient-reported outcomes than established VA measures both far and near [71].

Our team initially investigated the clinical utility of qCSF in a cohort of eyes with retinal vein occlusion (RVO). Relative to age-matched controls, eyes with RVO demonstrated statistically significant reductions in all outputs of the qCSF, including AULCSF, contrast acuity, and contrast sensitivity thresholds at all spatial frequencies [75]. In the subgroup analysis of the RVO eyes with VA of 20/30 or better, contrast sensitivity was still found to be significantly reduced compared to controls as measured by AULCSF, contrast acuity, and contrast sensitivity thresholds at 3, 6, 12, and 18 cpd. Of note, in a small sample of five eyes that had resolution of macular edema following anti-VEGF injections, there was a marked improvement in AULCSF compared to change in VA (0.523 log units in CSF vs. 0.144 log units in the VA) [75]. Our team followed the above study with a structural–functional association study, showing that the extent of disorganization of the retinal inner layers (DRIL) is not only associated with worse VA but is also significantly associated with reduced contrast sensitivity in eyes with history of macular edema secondary to RVO [76].

Our team has also previously reported on the qCSF method in eyes with macula-off rhegmatogenous retinal detachment (RRD) after surgical repair. Eyes with a history of macula-off RRD had statistically significant reductions in AULCSF and in contrast sensitivity thresholds across all spatial frequencies compared to healthy age-matched control eyes [74]. Similarly, in the subgroup analysis of the macula-off RRD eyes with VA better than or equal to 20/30, contrast sensitivity function was still significantly reduced compared to fellow eyes and controls. This may explain why patients after RRD continue to have visual complaints, despite the great anatomic success in reattachment surgery [74].

In our recently published work [79], our team has also examined a large cohort of 151 patients who retained good visual acuity and with different macular diseases including RVO, history of macula RRD, and dry and wet AMD. Similarly, reductions in AULCSF and contrast thresholds at all spatial frequencies were found to be significantly reduced in eyes with various macular diseases with VA as good as 20/30, and even in the eyes with VA ≧ 20/20^−1^, compared to 93 healthy control eyes. 

#### 2.2.4. The Premise of the qCSF in Non-Neovascular Age-Relate Macular Degeneration 

Preliminary unpublished data on a large cohort of dry AMD patients at our institution suggests that CSF as measured by the qCSF is significantly decreased in early dry AMD compared to healthy controls, despite no differences in VA (Figure 3). Further, the qCSF seems to be able to differentiate between dry AMD stages, hence emerging as a promising functional outcome metric to monitor disease progression. 

The currently ongoing MACUSTAR Consortium seeks to investigate multiple functional, structural, and patient-reported potential endpoints in intermediate AMD [80]. Contrast is planned to be tested using the Pelli-Robson chart, hence tested at only one spatial frequency (0.5–1 cpd) [81]. The Food and Drug Administration (FDA) mandates that contrast sensitivity can only be used as a functional endpoint in clinical trials if significant differences are reported in at least two spatial frequencies, when tested with the Pelli-Robson, and contrast sensitivity does not fulfil the FDA criteria as a clinical trial endpoint [81,82]. This further emphasizes the currently unmet need for a contrast sensitivity test that could meet the FDA’s prerequisites so that contrast sensitivity will be eventually leveraged as a functional endpoint in future clinical trials, as an independent parameter for determining the degree of visual impairment, disease progression and potential treatment effects of experimental new treatments for non-neovascular AMD.

#### 2.2.5. The Premise of the qCSF Method in Neovascular Age-Related Macular Degeneration 

In the field of neovascular (wet) AMD, the use of contrast sensitivity function as a more sensitive functional outcome may demonstrate larger treatment effects after therapeutic interventions such as anti-VEGF injections compared to VA, larger difference in the treatment effect of different therapeutic interventions and even treatment effect when VA improvement may have plateaued. Figure 4 shows a representative example of a patient with wet AMD with OCTs before (A) and after (B) the injection of an anti-VEGF agent with resolution of intraretinal and sub retinal fluid, where the treatment effect measured with contrast sensitivity is many times larger than the treatment effect measured with VA. Specifically, in this example, the improvement in VA (ΔVA) is only 0.1 LogMAR (1 line), from 0.40 LogMAR (20/50) to 0.30 LogMAR (20/40), yet the improvement in contrast sensitivity thresholds (Δcs) is many times larger, 0.46 LogCS at 3 cpd and 0.55 LogCS at 6 cpd. (Figure 4).

Current trials for combination therapies for neovascular (wet) AMD face the challenge of measuring the superiority of a combination effect, relative to monotherapies that provide a strong comparator signal with high efficacy. There is a critical unmet need for a visual function endpoint that can provide the sensitive and precise signals required to initiate and track neovascular AMD treatment over time. Such an endpoint would provide the valuable data about neovascular AMD therapies needed by regulators to determine their benefit-risk profile and by payors to evaluate their cost-effectiveness.

## 3. Conclusions

Quantifying meaningful changes in vision is constrained by what is clinically measurable at baseline, during disease progression, or after therapeutic intervention. As researchers work to develop new treatments for both non-neovascular and neovascular AMD, employment of better visual function metrics would aid in detecting subtle changes in visual function and evaluate disease progression and the effectiveness of novel therapeutics [18,72,83].

Compared to visual acuity, contrast sensitivity seems to correlate better with subjective visual impairment and vision-related quality of life, may be affected earlier in the course of neurodegenerative disorders [28,29,30] and may detect more subtle changes in visual function [24,31,32]. Yet the inherent imperfections of the currently available contrast sensitivity tests have prevented adoption of contrast sensitivity as a functional endpoint in clinical practice or in clinical trials. The Pelli-Robson chart only operates at one spatial frequency and present pre-printed charts with grating contrasts across pre-determined spatial frequencies are limited by poor range of resolution and low test–retest reliability. With the advent of the qCSF, contrast sensitivity can now be efficiently measured across multiple spatial frequencies with both high sensitivity in detecting subtle changes in visual function and great test–retest reliability, fulfilling the FDA criteria for functional endpoints [81,82] and emerging as a promising visual function endpoint in AMD, both in clinical practice and future clinical trials.

## Figures and Tables

**Figure 1 jcm-10-02768-f001:**
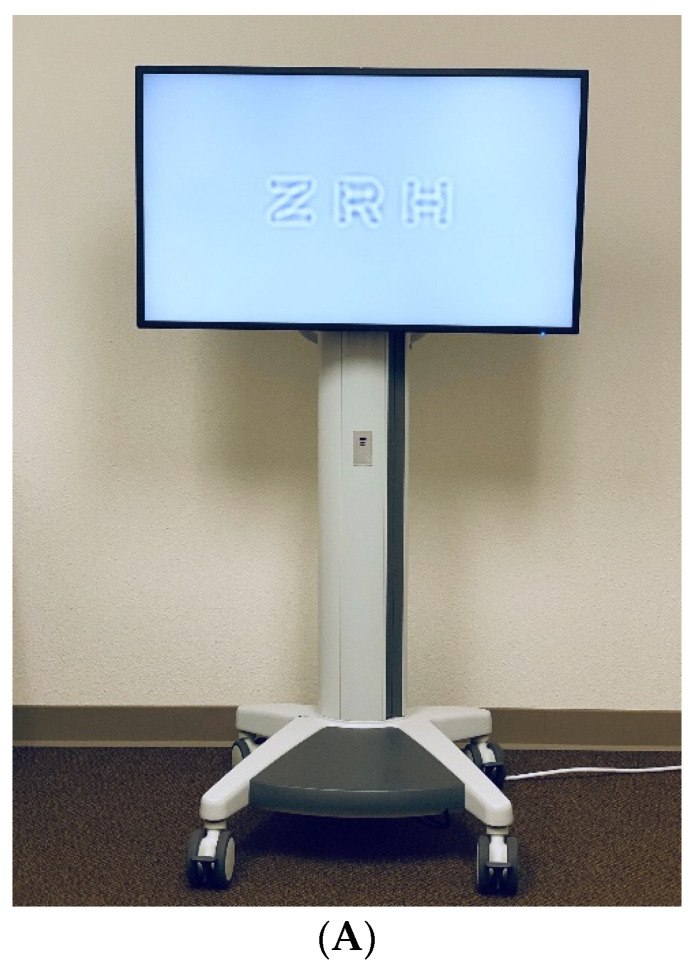
(**A**). The computerized AST Platform ut46” LED screen simultaneously displays three filtered Sloan letters of the same spatial frequency in a horizontal line with decreasing contrast. The contrast of the far right letter is chosen by the qCSF and is usually near threshold contrast, with the middle and left letters displayed at two and four times the contrast of the right letter, respectively (previously unpublished figure). (**B**). Screenshot of remote control tablet during a test session. The trial history (correct answers, triangles; mistakes, crosses; ’no answer’ responses, slashes) and the current best estimate of the CSF are shown in the top panel, with spatial frequency on the x-axis and contrast sensitivity on the y-axis. The current trial is highlighted in blue (previously unpublished figure).

**Figure 2 jcm-10-02768-f002:**
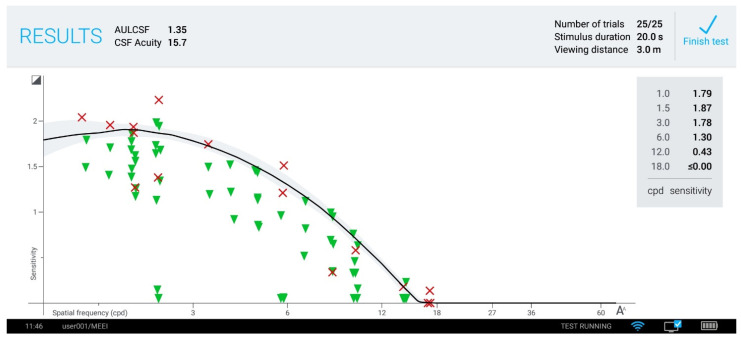
Screenshot of remote control tablet, showing test results after a session. The solid line denotes median sensitivity estimate for the possible CSFs, with the shaded region denoting the 66% confidence interval. Contrast acuity measures the smallest optotype visible with the highest level of contrast, as illustrated by the intersection of the contrast sensitivity function curve with the x-axis (previously unpublished figure).

**Figure 3 jcm-10-02768-f003:**
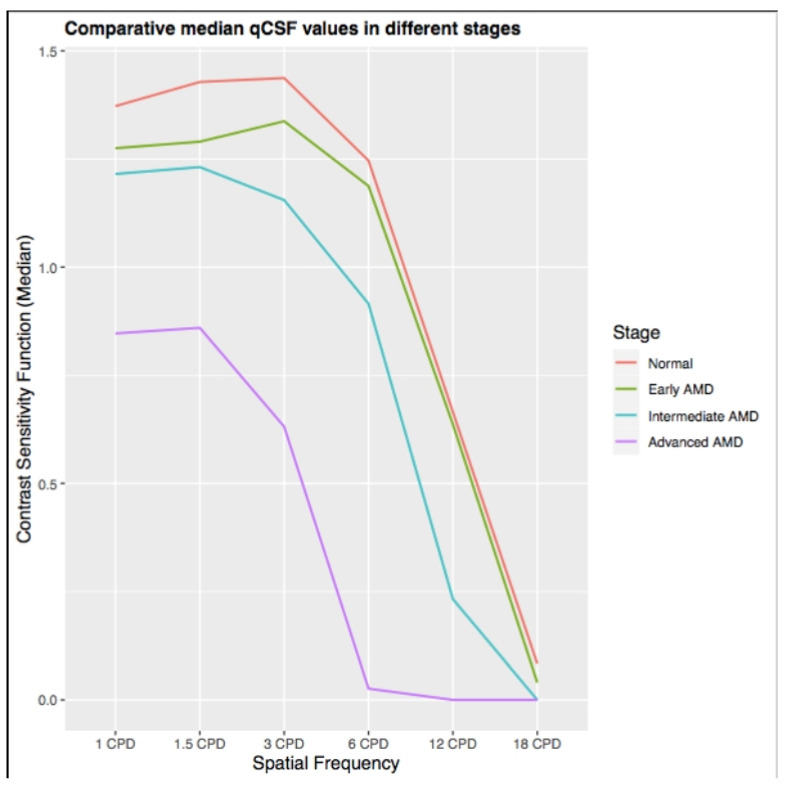
Contrast sensitivity function as measured by the qCSF in early, intermediate and advanced non-neovascular (dry) AMD compared to healthy controls. The qCSF seems to be able to differentiate between non-neovascular (dry) AMD stages (previously unpublished figure).

**Figure 4 jcm-10-02768-f004:**
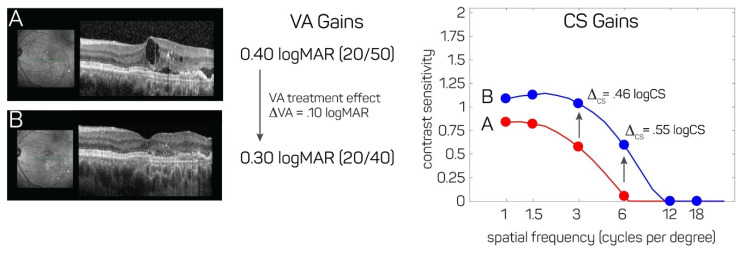
Representative example of a patient with neovascular (wet) AMD before and after an anti-VEGF injection, where treatment effect measured with contrast sensitivity is many times larger than treatment effect measured with visual acuity. On the left side of the figure are OCTs before (**A**) and after (**B**) the injection of an anti-VEGF agent showing resolution of intraretinal and sub retinal fluid. The improvement in VA (ΔVA) is only 0.1 LogMAR (1 line), from 0.40 LogMAR (20/50) to 0.30 LogMAR (20/40), yet the improvement in contrast sensitivity thresholds (Δcs) is many times larger, 0.46 LogCS at 3 cpd and 0.55 LogCS at 6 cpd (previously unpublished figure).

## Data Availability

Unpublished data, available upon request.

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
