# Peer review of "Measuring the Contrast Sensitivity Function in Non-Neovascular and Neovascular Age-Related Macular Degeneration: The Quantitative Contrast Sensitivity Function Test"

_jcm, 2021, doi:10.3390/jcm10132768_

Round 1

Reviewer 1 Report

Congratulations on an excellent article that helps clinicians in primary care as well as those working in research to better understand the importance of measuring contrast sensitivity. Real-world visual performance is always the most relevant metric for our patients and I am excited to see that this advanced technology enables efficient data collection and analyzation. I am hoping instrument design will enable easy incorporatation into the examination flow of primary eye care practices.

Author Response

Reviewer #1

1. Partially read lines are given a ‘ + ’ or ‘ - ’ designation, though there is no accepted standard on how this is recorded. 

Thank you for your suggestion, we edited accordingly adding the sentence: ‘Partially read lines are given a ‘ + ’ or ‘ - ’ designation, though there is no accepted standard on how this is recorded.’

2. Two lines on the P-R chart represents 0.3 logCS change; a 15 letter loss on a VA chart with 0.1 log steps represents a change of 0.3 logMAR. This does not present an argument in favor of the previous statement that the two are independent of each other. They are measuring different aspects of visual performance, but still show a relationship since CS will be lower with reduced VA. However, CS can also decline in advance of declines in high contrast VA. See https://doi.org/10.1002/mnfr.201801053 where the same article is referenced, but where a model is also presented to help clarify the impact a decline in CS may have on visual function when put in terms of VA changes that are more familiar to the clinician. Given the close association to real-world function, it is likely that the same log change has more real-world impact with CS change.

Thank you for your comment. In their article, Rubin et al do not claim that CS and VA are independent to each other, rather that contrast sensitivity seems to be an independent factor relative to VA (specifically) for self-reported visual disability. This is the claim we make in our manuscript, that CS and VA independently affect subjectively perceived visual impairment and vision-related quality of life, not that there is no correlation between them.

We read with interest the article by Roark and Stringham and the suggested model. Thank you for bringing this article to our attention. Albeit very informative, it offers no original data in a study cohort to support the suggested model. We certainly agree that ‘Further analysis shows that improvement in small-target VA moves the lower right portion of the CSF curve to the right, while enhancement in CS at intermediate target sizes elevates the highly sensitive central portion of the function’. Yet when it comes to quantifying the relationship between VA and CS and suggesting a model, recruiting a study cohort and employing a vision-related quality of life questionnaire to test that model along with VA and CS testing would be very meaningful to clinicians.

We are in complete agreement with reviewer suggesting that  ‘it is likely that the same log change has more real-world impact with CS change.’ This is exactly what we are aiming to illustrate by our sentence “a 6-letter loss of contrast sensitivity on the Pelli-Robson chart has a similar impact on self-reported visual disability as a 15-letter loss of visual acuity"

3. This is not actually correct. The Pelli-Robson letter size is fixed at 4.8mm, which is close to a VA of  20/650 at 1 meter, and approximately 20/220 at a 3 meter distance. This would equate to less than 1 cpd at 1 meter but spatial frequency becomes closer to 3 cpd at 3 meters. It is interesting that for normal subjects, results are similar between these test distances.

Thank you for the kind remark. Indeed it’s interesting that for patients who have good VA and CS, 1m vs 3m test distance seem to not affect the CS measured by the Pelli-Robson chart. Yet, this is not the usual clinical scenario in patients visiting a retina clinic.

The size of the letters presented by the Pelli-Robson’s is indeed around 20/650 at 1 meter, which is around 0.5-1 cpd; and indeed around 20/220 at a 3 meters which is approximately 3 cpd.

There are some studies that have used 3 meter testing distance, yet this is not a very common clinical scenario. The manual recommends testing at 1m meter, and this is how CS testing with the Pelli-Robson’s chart is typically performed in the clinical setting, only at 1 meter, hence the letter size is indeed close to 20/650; this in turn means it only tests contrast sensitivity at a spatial frequency between 0.5-1 cpd. And it would not be possible to alter the testing distance to test different spatial frequencies as the measurements at 1 and 3 meters are not affected by testing distance only in normal VA&CS participants.

To illustrate the above in a more comprehensive manner, we changed the sentence ‘All letters are of the same optotype size (20/63), hence by the Pelli-Robson clinicians only measure contrast sensitivity in one spatial frequency (0.5-1 cycles per degree)’ so that it now reads: All letters are of the same optotype size (4.8cm), and at the testing distance of 1 meter that the test’s manual suggests, this corresponds to a Snellen equivalent of approximately 20/650. Hence, employing the Pelli-Robson test, clinicians only measure contrast sensitivity at one spatial frequency, which in the typical clinical setting is between 0.5-1 cycles per degree.’

Reviewer 2 Report

Age-related macular degeneration (AMD) is a hot topic in the ophthalmology field, which affects hundreds of million peoples globally. There is no sufficient treatment for the advanced form of AMD, which emphasis the demand for diagnosis approaches for the early stage of AMD. In this review, Filippos and colleagues compiled the panels of valuable information on contrast sensitivity function (CSF) test, and its application on AMD diagnosis, and anti-VEGF therapy assessment of wet AMD. Overall, the manuscript is well-organized and informative. I only have few suggestions:

  1. Uniform the format of the figures. Add the resource of images in the figure legend (Unpublished or publication with copyright permission).
  2. Can the authors discuss a bit about chromatic and achromatic CSF in AMD?

Author Response

Reviewer #2

1. Uniform the format of the figures. Add the resource of images in the figure legend (unpublised or publication with copyright permission)

As per your suggestion, we have now uniformed the format of our figures. They are all unpublished figures hence no copyright permission is needed. We have added the sentence ‘(previously unpublished figure)’ at the end of all our figure legends, per your suggestion.

2. Can the authors discuss a bit about chromatic and achromatic CSF in AMD

Thank you for your suggestion. Chromatic contrast sensitivity is a distinct visual function metric that is beyond the realm of our current review, which is also the case for visual function metrics such as microperimerty, dark adaptation and ERG. Despite the commonality in the terminology between ‘contrast sensitivity’ and ‘chromatic contrast sensitivity’, they are two completely separate visual function metrics. To make that clear in our review, we have added the following sentence: 

‘Contrast sensitivity quantifies the lightness or darkness needed to identify a target against its

background. Chromatic contrast sensitivity will not be discussed herein, as it describes the ability to discriminate between stimuli based on their chromaticity difference alone, independent of any luminance contrast.’